# The Effect of Bean Seed Treatment with Entomopathogenic Fungus *Metarhizium robertsii* on Soil Microarthropods (Acari, Collembola)

**DOI:** 10.3390/insects13090807

**Published:** 2022-09-05

**Authors:** Tatiana Novgorodova, Natalia Vladimirova, Irina Marchenko, Tatyana Sadokhina, Maksim Tyurin, Lyudmila Ashmarina, Dmitry Bakshaev, Georgy Lednev, Viktor Danilov

**Affiliations:** 1Institute of Systematics and Ecology of Animals, Siberian Branch, Russian Academy of Sciences, Frunze 11, 630091 Novosibirsk, Russia; 2Siberian Federal Scientific Center of Agro-BioTechnologies, Russian Academy of Sciences, Central 2b, 630501 Krasnoobsk, Russia; 3All-Russian Institute of Plant Protection, Podbelskogo 3, 196608 St. Petersburg, Russia

**Keywords:** agricultural crops, endophytic systems, entomopathogenic fungi, soil microarthropods, community structure, *Vicia faba*

## Abstract

**Simple Summary:**

Plant pest control is essential for agriculture. The treatment of agricultural crops with entomopathogenic fungi protects plants against pests and can also increase plant growth, but at the same time may disturb the community structure of non-target beneficial invertebrates, in particular soil microarthropods. This can have a negative impact on soil fertility and ultimately on crop yields. The effect of the treatment of broad bean (*Vicia faba* L.) seeds, with the entomopathogenic fungus *Metarhizium robertsii*, on the abundance and structure of soil microarthropod communities in the rhizosphere was assessed. Compared with the control, no adverse effect was revealed on both the abundance of soil microarthropods, including mites (Mesostigmata, Oribatida, Astigmata, Prostigmata) and springtails (Collembola), and the structure of microarthropod communities. Therefore, dressing seeds with a conidial suspension for plant inoculation, with entomopathogenic fungi (at least *M. robertsii*), can be assumed as a potentially safe method of plant protection for soil microarthropods.

**Abstract:**

The treatment of agricultural crops with entomopathogenic fungi may disturb the structure of soil microarthropod communities, which can have an adverse impact on soil fertility and, ultimately, on the yield. The effect of the treatment of broad bean (*Vicia faba* L.) seeds, with the entomopathogenic fungus *Metarhizium robertsii*, on the abundance and community structure of soil microarthropods in the rhizosphere was assessed in different phases of plant vegetation in a two-year experiment. Under the conditions of the gradually decreased abundance of *M. robertsii* both in the soil cenoses and in the plants during summer, no adverse effect was revealed of the bean seed treatment, with the entomopathogenic fungus, on the abundance of soil microarthropods (Acari: Mesostigmata, Prostigmata, Oribatida and Astigmata; Collembola) and the structure of their communities. Similar results were obtained in the analysis, taking into account the positive colonization of plants. Some changes in the microarthropod community structure were explained primarily by the spatial heterogeneity of the field, the hydrothermal regime, and the features of the microarthropod life cycles. The results indicate the possibility of using dressing seeds with conidial suspension for plant inoculation with entomopathogenic fungus (at least *M. robertsii*) as a potentially safe plant protection method for non-target soil microarthropods.

## 1. Introduction

Various microorganisms are able to colonize plants and form stable endophytic systems [1]. Among all endophytic microorganisms, the entomopathogenic filamentous fungi from the genera *Beauveria*, *Metarhizium*, *Cordyceps* (=*Isaria*), and *Akanthomyces* (=*Lecanicillium*) are of particular interest [2]. A large number of papers report discovering these micromycetes in both cultivated and wild plants. It was found that these fungi can have an adverse effect on both phytophagous insects and phytopathogens (bacteria and fungi), and in some cases, they can enhance the growth of plants and increase their adaptability to adverse abiotic condictions [2,3,4]. Today, a promising direction in plant protection is the use of entomopathogenic fungi as producers of multifunctional biological products that not only reduce the number of harmful arthropods but also have a significant adverse impact on phytopathogens, as well as growth-stimulating and stress-resistance effects on the protected crop. Among the entomopathogenic fungi as producers of mycoinsecticides, representatives of the genera *Metarhizium* and *Beauveria*, which account for more than 70% of commercial products, are of the greatest interest [5]. These fungi can spread not only through insects, soil, and other substrates but can also form stable endophytic systems. Entomopathogenic fungi are often recorded in the soil, including the rhizosphere of plants, where they can markedly affect various soil invertebrates [6], primarily soil microarthropods (mites and springtails), which, in turn, are one of the key functional elements of natural and anthropogenic landscapes [7,8]. Microarthropods play an important role in soil formation, accelerating the decomposition and humification of organic residues [9,10,11]. It is known that soil microarthropods are closely associated with fungi and can form associations [12]. On the one hand, microarthropods can significantly affect the species composition and structure of microbial communities that are formed during the decomposition of plant residues, as well as the growth and metabolic activity of microorganisms [13]. Microarthropods can be involved in the spread of fungi in the soil and provide optimal conditions for their growth, in particular, the entomopathogenic fungi (e.g., *Beauveria bassiana*) [14]. In turn, various microorganisms play an important role in the life of microarthropods, including their distribution [13]. Bacteria, as well as fungal spores and hyphae, form the basis of the diet of groups such as Astigmata, Oribatida, Prostigmata, and Collembola [8,15]. Along with other members of the soil fauna, microarthropods can suppress the abundance of pathogenic species of *Fusarium* fungi in agrocenoses, shortening the length of hyphae of different *Fusarium* species and increasing the rate of decomposition of mycotoxins [16,17,18,19]. At the same time, soil microarthropods, due to their soft integuments, are an excellent target for entomopathogenic fungi [20,21,22], which are present in the soil [23]. It is known that microarthropods can carry spores of entomopathogenic fungi on the surface of their bodies, thereby horizontally transmitting infection in soil insect communities. In particular, this was shown for different mites [24], as well as for collembolans [25,26].The causative agents of mycoses are known to be destructive to members of most groups of soil microarthropods, in particular Astigmata, Oribatida, Prostigmata, and Mesostigmata mites [27,28]. This raises the question of the susceptibility of microarthropods to transmitted fungal infections. There is no consensus on this issue.

Treatment of agricultural crops with entomopathogenic fungi may disturb the structure of soil microarthropod communities. This, in turn, can have a negative effect on soil fertility and ultimately on yields. The aim of this study was to assess the effect of the treatment of broad bean (*Vicia faba* L.) seeds with the entomopathogenic fungus *Metarhizium robertsii* on the composition and structure of soil microarthropod communities in the rhizosphere.

## 2. Materials and Methods

Field experiments were carried out in 2019 and 2020 on the field of the experimental station of the Siberian Institute of Forage of the Siberian Federal Scientific Centre of Agro-BioTechnologies, Russian Academy of Sciences, located in the northern forest-steppe of the Ob region, Novosibirsk Region, Russia. The soil type in this region is leached chernozem, medium thick, medium loamy, with the organic carbon content in the soil 3.48%, pH 5.3. The amount of absorbed bases is 58–61 mg/eq. per 100 g of soil. The predecessor was fallow land.

### 2.1. Fungus and Plants

In the experiments, the fungus *M. robertsii* strain P-72 was used. It was obtained from the collection of microorganisms of the Institute of Systematics and Ecology of Animals, Siberian Branch, Russian Academy of Sciences (Novosibirsk). The fungus species was identified by the sequence of the translation elongation factor region (EF1 α). Seeds of broad bean cultivar Sibirskie were used for inoculation.

The conidial mass was accumulated by the two-phase cultivation method. First, a submerged fungal culture was accumulated on Sabouraud’s medium in an incubator shaker at 160 rpm and 26 °C for 4 days. Then, the submerged culture (5 mL) was inoculated into Petri dishes (diameter—15 cm) containing twice-autoclaved seeds of millet. Double autoclaving ensured the sterility of the substrate. The dishes were incubated at 24–25 °C in the dark for 10 days. Thus, obtained culture was dried at 25 °C and 18% RH for 10 days and then homogenized with a ball mill. The titer of 1 mg of *M. robertsii* conidia in 1 mL of water was 5 × 10^6^ conidia/mL.

Fungal conidia were suspended in a Tween 20 solution (0.04%) at a concentration of 5 × 10^7^ conidia/mL. Using this concentration of fungal conidia suspension is optimal, since it makes it possible to obtain a positive effect on plant growth, reduces the infection level of the seed material with phytopathogens, and significantly decreases the development and prevalence of root rot disease [4,29,30,31,32].

The broad bean seeds were treated with suspension and allowed to dry immediately before sowing. The control variant was treated with Tween 20 (0.04%). The introduced volume of the suspension was 2.5 L per 20 kg grains of broad beans. Treatment of planting material by dressing seeds with conidial suspension is a typical method used for plant inoculation [4,29,30,31].

In both cases, broad beans were sown on 16 May 2019 and on 19 May 2020, when the soil temperature at a depth of 6–8 cm reached 8–10 °C, in wide rows (×70 cm), at a seeding rate of 400 thousand viable grains per hectare. The plot length in the experiment was 10 m, the width was 3.9 m, and the plot area was 39 m^2^.

To assess plant colonization by the entomopathogenic fungus, the research team used a modified Sabouraud medium was used with an antibiotic cocktail as described by Tomilova et al. [29]. The middle part of the root, the lower third of the stem and the leaf from the middle plant layer were selected for analysis. These plant parts were washed with tap water and sterilized with 0.5% sodium hypochlorite and 70% ethanol as described by Parsa et al. [33]. The plant parts were imprinted on the medium and then placed on its surface in 90 mm Petri dishes. The percentage of fungus-positive plant parts was calculated. Samples showing fungal growth on the imprints were excluded from the analysis.

To assess rhizosphere colonization, the pre-washed roots of the plants were plated without surface sterilization. After preliminary washing of a dense layer of basal soil on bean plants (at least twice), the roots were washed three times for 1 min using Vortex-Mixer PV-1 (Grant-bio) and plated on the medium in Petri dishes.

The dishes containing the plant and soil samples were incubated at 24 °C for 14 days, followed by the detection of fungal growth from the plant organs. *Metarhizium* fungi were detected visually and by light microscopy. Colony-forming units in the rhizosphere soil were quantified according to the methods described earlier [29].

### 2.2. Experiment Design

The field experiment included the following treatments: control (1) and *M. robertsii* (2). The variants in the experiment were arranged systematically, in four (2019) and five (2020) replicates. In each case, during the entire field season, soil temperature and relative humidity were recorded with temperature and humidity sensor data loggers (TR-2V), which were placed at four points along the diagonal of the experimental field at a depth of 5 cm. The average daily temperature and humidity data are summarized in Table 1.

### 2.3. Material Collection

The soil substrate was collected and the microarthropods were extracted from the samples and fixed according to the standard methods [34]. Soil samples were taken at the base of plants with a standard cylindrical drill (125 cm^3^) at a depth of 5 cm (10 samples per plot). Microarthropods were extracted in laboratory conditions using modified Berlese funnels without additional sources of light and heat. This method of microarthropod extraction provides natural conditions for soil drying and avoids or at least minimizes the possibility of missing some individuals due to the expedited drying process. Extraction continued for 10–15 days until the soil samples became completely dry.

To test the uniformity of conditions (the composition and structure of soil microarthropod communities), 10 randomized samples were taken from the experimental plots on 8 May 2019 and on 12 May 2020 immediately before plowing and sowing seeds.

In the course of experiments, the abundance of microarthropods was determined three times per season during different periods of the plant growing season: the branching phase (11 June 2019; 3 June 2020; hereinafter, for short, June), the budding phase (23 July 2019; 15 July 2020; hereinafter, July), and the seed maturation phase (30 August 2019; 17 August 2020; hereinafter, August). In total, 560 soil samples were collected (250—in 2019, 310—in 2020), from which more than 48,000 microarthropod specimens were ousted (39,135—in 2019, 9537—in 2020). The density of microarthropods in the soil horizon 0–5 cm was recalculated per 1 m^2^ by multiplying by the following coefficient: 10.000 cm^2^/19.625 cm^2^ (the sampler base area).

Microarthropods were analyzed and identified using the following microscopes: Altami CM0745-T, Hund wetzlar SM33, Axioscop 40 (Zeiss), and Primo Star (Zeiss). To determine the taxonomic affiliation, slides of microarthropods were prepared using Hoyer’s medium. The general system of microarthropods is given according to Lindquist et al. [35] and Zhang [36]. Since a number of authors still consider Endeostigmata as part of the order Prostigmata [37,38,39], and also due to the low abundance of both groups in the samples, these groups were considered in a single block as Prostigmata.

The materials are stored in the collection of the Siberian Zoological Museum, Institute of Systematics and Ecology of Animals, Siberian Branch, Russian Academy of Sciences, Novosibirsk.

### 2.4. Data Analysis

To obtain preliminary information on the distribution of microarthropods (both in general and their individual groups, ind./m^2^) in the field before the start of the experiments, the Shapiro–Wilk test was used. *p* values less than 0.05 meant uneven distribution. Because of the non-normal distribution of microarthropods in most cases (Shapiro–Wilk test, *p* < 0.05), the influence of plant seed treatment on both the abundance of microarthropods and the structure of their communities (the proportion of individual groups) was estimated using the non-parametric Mann–Whitney test. To exclude the effect of the low level of fungal colonization of bean plants, in addition to the general data analysis, a further analysis was carried out taking into account positive plant colonization by the fungus *M. robertsii* in 2020. Only colonized plants in the *M. robertsii* treatment and uncolonized plants in the control were analyzed. This approach was used to analyze the data obtained only in June and July, because of the absence of confirmed colonization of plants examined in August [40]. Data were statistically processed using the STATISTICA v.8.0.725 and Microsoft Excel software.

## 3. Results

The analysis results on the presence of entomopathogenic fungus *M. robertsii* in the soil of the bean root zone, as well as the effects on plants, have been reported previously in the previous paper [40]. Briefly, in 2019, the level of colonization in the plots treated with *M. robertsii* was quite low (0–2.5% *Metarhizium*-positive plants) and did not differ significantly from the control. In 2020, *M. robertsii* successfully colonized plants and accumulated in the aboveground and underground organs, as well as in the rhizosphere. However, the entomopathogenic fungus was isolated most actively from the young plants in the initial stages of the growing period, after which the density of entomopathogenic fungi in the rhizosphere zone decreased in the course of the plant growing season to the control level [40]. Despite the low level of plant colonization by *M. robertsii* in 2019, the treatment with *M. robertsii* significantly reduced the development and prevalence of root rot disease in both 2019 and 2020 [40]. Moreover, in both cases, strong positive effects of *M. robertsii* treatments on plant height were observed [40].

Before the start of the experiments (immediately before ploughing the field) in May 2019, microarthropods of five groups were identified in soil samples: mites (Mesostigmata, Prostigmata, Oribatida and Astigmata) and springtails (Collembola). The distribution of the population density of both microarthropods in general and individual groups turned out to be rather uneven (Shapiro–Wilk test: microarthropods in total, W = 0.838, *p* = 0.042; Mesostigmata, W = 0.731, *p* = 0.002; Prostigmata, W = 0.50, *p* < 0.0001; Oribatida, W = 0.778, *p* = 0.008; Astigmata, W = 0.532, *p* < 0.0001; Collembola, W = 0.730, *p* = 0.002). In May 2020, members of four groups of microarthropods were identified in soil samples: mites (Prostigmata, Oribatida and Astigmata) and springtails (Collembola). Mesostigmata were absent in the samples. The distribution of both microarthropods in total and individual identified groups was fairly uniform (Shapiro–Wilk test: microarthropods in total, W = 0.999, *p* = 0.963; Prostigmata, W = 0.964, *p* = 0.637; Oribatida and Collembola, W = 1, *p* = 1; Astigmata, W = 0.999, *p* = 0.961).

In the summer period from June to August in both 2019 and 2020, members of five groups of microarthropods were identified in the samples: mites (Mesostigmata, Prostigmata, Oribatida and Astigmata) and springtails (Collembola).

Treatment with *M. robertsii* had no significant effect on the abundance of microarthropods in most cases. The significant differences were found only for total abundance of microarthropods in June 2019, which was significantly lower in the plots with *M. robertsii* treatment than in the control (Figure 1). At the same time, the opposite situation was observed in August: a non-significant increase in the total abundance of microarthropods in the plots with *M. robertsii* treatment compared to the control (Figure 1). Comparative analysis of data for individual groups of microarthropods revealed no significant differences both in 2019 and 2020. However, a slight (but non-significant) increase in population density was noted for Astigmata in August 2019 when treated with the fungus *M. robertsii* (Table 2 and Table 3).

To exclude the effect of low/-un- colonized bean plants in the treatment, additional analysis was carried out in 2020 on positive colonized plants of the treatment vs un-colonized plants of the control. There were no significant differences in the abundance both in microarthropods in total and individual groups (Table 4).

The ratio of the identified groups of microarthropods (Acari: Mesostigmata, Prostigmata, Oribatida, Astigmata; Collembola) in the community changed in the course of the growing season and had certain peculiarities. In 2019, at the beginning of the season, the microarthropod community was dominated by Oribatida (51%). The proportion of Astigmata in June was 19%, Mesostigmata—15%. Other groups did not exceed 10%: Collembola—10%, Prostigmata—5%. In July, the abundance of Astigmata highly increased and made up 98% of the total population of microarthropods. In August, due to an increase in the total abundance of other groups of microarthropods (Collembola, Oribatida), the percentage of Astigmata slightly decreased and amounted to 57%. The proportion of Prostigmata in the community throughout the season did not exceed 5%.

Similar results were obtained in 2020. At the beginning of the season, the microarthropod community was dominated by Oribatida (38%). The proportion of Astigmata in June was 27%. Later, the proportion of Astigmata gradually increased in the course of the growing season and in August reached 89%, whereas the proportion of other groups (Prostigmata, Oribatida and Mesostigmata) gradually decreased, and by the end of the season accounted for only 1–2%. The proportion of Collembola in the community was highest in July (52%), whereas in June and August it was only 4% and 7%, respectively.

There were no significant differences in the proportion of individual microarthropod groups in the community, between the treatment and the control in both 2019 and 2020 (Table 5 and Table 6). The comparative analysis of the data (2020), taking into account the positive colonization of plants by *M. robertsii*, also revealed no significant differences, between the treatment and the control in the proportion of individual groups in June and July (Table 7).

## 4. Discussion

At the first step of this complex study, it was shown that, after inoculating bean seeds with *M. robertsii*, the fungus could more or less successfully colonize plants, accumulating in their aboveground and underground organs, as well as in the rhizosphere, a typical trophic site of soil microarthropods [40]. The development and prevalence of root rot disease was found to be significantly lower in the plots with fungus bean seed treatment in both 2019 and 2020. Moreover, treatment by dressing broad bean seeds with a suspension of *M. robertsii* was found to have a significant positive effect on plant growth, even at a low level of plant colonization by the fungus in 2019 [40], which can also have a direct or indirect effect on the abundance and community structure of microarthropods. It should be noted that, in the course of the growing season, this fungus is gradually eliminated both in the soil cenoses and in the plants themselves [40].

The communities of soil microarthropods in the study area included mites (Mesostigmata, Prostigmata, Oribatida, and Astigmata) and springtails (Collembola), which are common both in natural biocenoses and in agrocenoses [8,41,42] and represented by groups with different dietary patterns playing important roles in biocenoses. There are saprophages and mycophages among Oribatida [43,44]; saprophages as well as consumers of fungi and bacteria among Astigmata [45]; predators, phytophages, saprophages, mycophages, and parasites among Prostigmata [39,46]; and active predators among Mesostigmata [47,48]. Collembolans feed on dead organic matter, bacteria, hyphae and spores of fungi, and algae, and promote the dispersal of microflora in the soil and plant litter [49,50].

The treatment of *V. faba* seeds with the entomopathogenic fungus *M. robertsii* in general had no significant effect on the structure of the microarthropod community, as well as on the abundance of individual groups and microarthropods in total. The only exception was found in June 2019 when the abundance of microarthropods in total was significantly lower in the plots with *M. robertsii* treatment compared to the control. However, no significant differences were found for the individual groups. Moreover, in August, due to the rapid increase in the number of Astigmata, a slight increase was noted in the total abundance of microarthropods in the plots with seed treatment with *M. robertsii* compared with the control. In general, the nature of slight changes in the abundance and structure of microarthropod communities corresponded to natural seasonal rhythms and is explained by the characteristics of the individual group’s biology, as well as by the spatial heterogeneity of the field, the hydrothermal regime, and the reserves of organic residues in the field [8,10,38,43].

Microarthropod communities were dominated by Oribatida at the beginning of the season, Collembola or Astigmata in the middle of the season, and Astigmata at the end of the season. This is due to the peculiarities of their life cycles. The majority of microarthropod groups (Astigmata, Prostigmata, Mesostigmata, and Collembola) have short life cycles and high reproduction rates. In Astigmata, development proceeds very quickly (5 days) under favorable conditions (humidity: 85–90%; temperature: 25–30 °C) and may take 28 days when the temperature drops to 16 °C [51,52]. In Mesostigmata, the life cycle is also temperature-dependent and usually lasts 6–7 days at 22–24 °C [47,53,54]. Collembolans can rapidly increase their density in places with high humidity and an abundance of food [53]. This was reflected in the lowest abundance of these groups in June. With an increase in average daily temperatures in the upper soil layers in the middle of the season (Table 1), the abundance of Astigmata and Collembola drastically increased. The abundance of both Mesostigmata and Prostigmata also increased in the middle of the season; however, the increase was not pronounced, and their proportion in the community was 2–6 times lower than the proportion of the first two groups in the middle of the season. Unlike other groups, oribatid mites are typical k-strategists: this group is characterized by a longer life cycle and developmental period, as well as a low reproduction level [43,44]. The development period of Oribatida ranges from several months to more than one year (favorable temperatures: 20–30 °C). This fact explains the highest density of their population and the proportion in the community at the beginning of the season (Table 2, Table 3, Table 5 and Table 6) at the lowest temperatures (Table 1). The abundance of Oribatida remained practically the same throughout the season; as a result, their proportion in the community sharply decreased from June to August due to the dominance of other groups.

Overall, environmental variables, primarily temperature and precipitation, can impact both plant colonization by entomopathogenic fungi and abundance of soil microarthropods. Nevertheless, results under variable plant colonization and positive plant colonization by fungi (colonized plants in *M. robertsii* treatment vs. uncolonized plants in the control), appeared to be consistent. 

Some studies showed that entomopathogenic fungi are fatal for members of most groups of soil microarthropods, in particular, for the mites Astigmata, Oribatida, Prostigmata, and Mesostigmata [24,27,28]. A long-term (14 days) contact of collembolans with sphagnum inoculated with these fungi resulted in death from mycosis [55]. However, other studies indicate negligible effect of entomopathogenic fungi to non-target soil microarthropods. A study of the effect of the fungus *M. brunneum* on non-target organisms showed that *Gaeolaelaps aculeifer* mites were not susceptible to infection [26]. Exposure of three species of collembolans (*Folsomia fimetaria* (L.), *Hypogastrura assimilis* (Krausbauer), and *Proisotoma minuta* Tullberg) to *Baueveria* and *Metarhizium* fungi by direct dipping of collembolans in a fungal suspension did not lead to their infection [26], although in target insects this method of infection leads to a significant mortality rate. Broza et al. [56] showed that fungal spores could attach to the cuticle surface of collembolans and form growth tubes, as well as exhibit enzymatic activity on the cuticle surface; however, no further disease progression was observed. The resistance to entomopathogenic fungi may reflect the adaptation of soil microarthropods to their habitats, because they evolved together with fungi as part of the soil community. 

The resistance to these micromycetes allows microarthropods to feed on them, otherwise, they might be potential pathogens. In particular, using modern molecular genetic methods, it was shown that, under natural conditions, in the absence of an artificial infectious background, collembolans carried spores of entomopathogenic fungi both in the gut and on the body surface [57,58]. In the above-mentioned study by Broza et al. [56], it was shown that collembolans could actively feed on the fungi *B. bassiana* and *M. anisopliae* (mycelium and conidia) and even increase their fertility while after passing through the digestive system of these microarthropods, the conidia of entomopathogenic fungi partially retained their viability (50–70%) and virulence for insects (e.g., ants). According to Dromph and Vestergaard [26], feeding of collembolans on entomopathogenic fungi contributes to the active spread of the latter in the soil cenoses, indicating a relationship between the groups that can be close to mutualistic. As a result of this type of relation, at the beginning of the season, a period with fairly low temperatures of the upper soil layer that is apparently characterized by the deficiency of natural fungi, seed treatment with *M. robertsii* can lead to the formation of “food spots” in the rhizosphere of young beans, which can be attractive to mycetophages, including Collembola, and promote their rapid development and reproduction and, possibly, redistribution within the plot. However, in the present study, no positive effects of the seed treatment with *M. robertsii* were detected.

In conclusion, no adverse effects of the bean seed treatment with the entomopathogenic fungus *M. robertsii* on the abundance and community structure of soil microarthropods were revealed. The nature of some changes in the structure of microarthropod communities during the experiment can be explained primarily by the spatial heterogeneity of the field, the hydrothermal regime, and the features of the life cycles of the microarthropods themselves. This indicates the possibility of using dressing seeds with a conidial suspension for plant inoculation with entomopathogenic fungi as a potentially safe plant protection method for non-target soil microarthropods.

## Figures and Tables

**Figure 1 insects-13-00807-f001:**
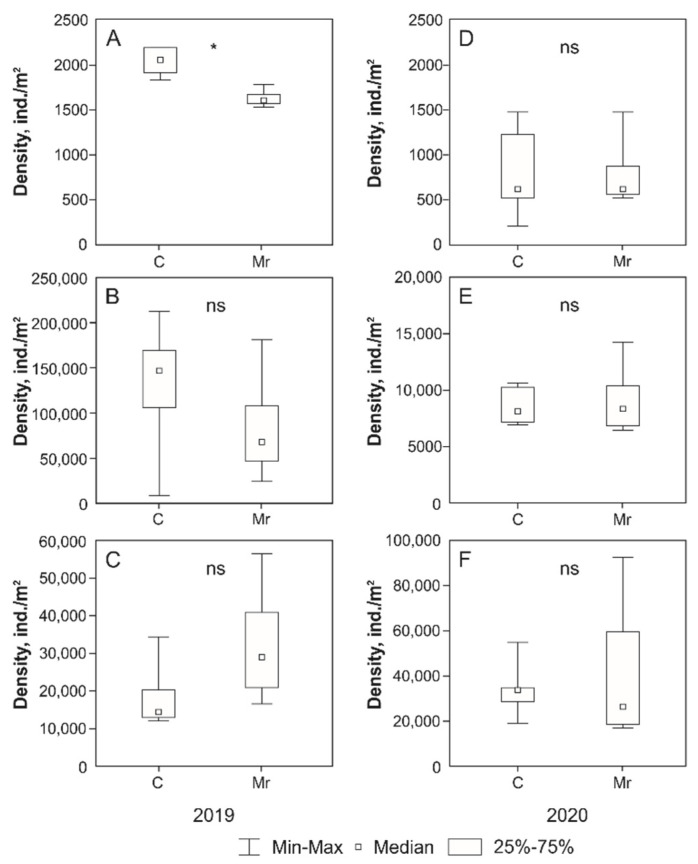
For 2019 and 2020. (**A**,**D**)—June; (**B**,**E**)—July; (**C**,**F**)—August. Mann–Whitney test: *—differences are significant (*p* < 0.05), ns—non-significant differences (*p* > 0.05).

**Table 1 insects-13-00807-t001:** Average daily temperature (T) and humidity (H) of soil (mean ± SD) in the experimental plots in the intervals between material collection in 2019 and 2020.

Date	2019		2020	
	T (°C)	H (%)	T (°C)	H (%)
8 May–11 June	15.00 ± 0.13	96.01 ± 2.07	-	-
12 May–03 June	-	-	19.52 ± 0.25	88.13 ± 3.30
12 June–23 July	20.54 ± 0.74	99.31 ± 0.67	-	-
4 June–15 July	-	-	20.54 ± 0.43	98.55 ± 1.58
24 July–30 August	20.20 ± 0.42	98.52 ± 1.41	-	-
16 July–17 August	-	-	22.23 ± 0.48	98.13 ± 1.78

**Table 2 insects-13-00807-t002:** Results of comparative analysis of the density of soil microarthropods (ind./m^2^) in the plots with different types of seed treatment in 2019 (Mann–Whitney test).

Group	Month	Treatment		U	*p*
		C	Mr		
Collembola	June	153 [140; 204] *	153 [153; 179]	7.0	0.77
	July	255 [230; 587]	281 [140; 727]	7.5	0.89
	August	2346 [1556; 6286]	5534 [5024; 7178]	4.0	0.25
Oribatida	June	1097 [867; 1288]	842 [689; 982]	4.5	0.31
	July	867 [587; 1173]	714 [434; 1122]	7.0	0.77
	August	2448 [2104; 2588]	816 [561; 3175]	7.0	0.77
Mesostigmata	June	230 [140; 344]	255 [153; 370]	7.0	0.77
	July	332 [204; 727]	179 [140; 242]	5.0	0.39
	August	1148 [1007; 1517]	1173 [892; 1326]	5.0	0.39
Astigmata	June	459 [319; 574]	255 [191; 332]	4.5	0.31
	July	142,366 [102,816; 165,903]	65,943 [43,886; 105,851]	6.0	0.56
	August	8211 [6184; 10,417]	20,349 [11,284; 28,815]	1.5	0.06
Prostigmata	June	102 [89; 115]	51 [51; 89]	5.5	0.47
	July	383 [191; 650]	51 [0; 217]	5.0	0.39
	August	510 [319; 1428]	204 [115; 816]	5.0	0.39

* Data are shown as ME [25; 75], where ME—median; 25 and 75—first and third quartiles, respectively. C—control, Mr—*Metarhizium robertsii*.

**Table 3 insects-13-00807-t003:** Results of comparative analysis of the density of soil microarthropods (ind./m^2^) in the plots with different types of seed treatment in 2020 (Mann–Whitney test).

Group	Month	Treatment		U	*p*
		C	Mr		
Collembola	June	0 [0; 102] *	0 [0; 51]	10.5	0.75
	July	4284 [3060; 4488]	4335 [3315; 5355]	11.0	0.84
	August	1989 [1938; 2652]	3111 [2040; 3519]	8.5	0.47
Oribatida	June	204 [ 204; 306]	357 [357; 408]	5.5	0.18
	July	204 [102; 306]	153 [153; 306]	10.5	0.75
	August	102 [51; 204]	204 [153; 204]	8.5	0.47
Mesostigmata	June	102 [51; 102]	51 [0; 51]	5.0	0.14
	July	867 [612; 969]	510 [408; 816]	6.0	0.21
	August	714 [612; 816]	714 [561; 918]	11.0	0.84
Astigmata	June	153 [102; 306]	204 [102; 306]	11.5	0.92
	July	1683 [1275; 2397]	2142 [1887; 3876]	11.0	0.84
	August	27,744 [24,378; 31,110]	20,961 [14,943; 55,029]	11.0	0.84
Prostigmata	June	51 [51; 102]	0 [0; 51]	8.0	0.40
	July	918 [867; 1020]	663 [612; 969]	9.5	0.60
	August	561 [408; 663]	357 [153; 510]	7.5	0.35

* Data are shown as ME [25; 75], where ME—median; 25 and 75—first and third quartiles, respectively. C—control, Mr—*Metarhizium robertsii*.

**Table 4 insects-13-00807-t004:** Results of comparative analysis of the density of soil microarthropods (ind./m^2^) in the plots out taking into account only colonized plants in *M. robertsii* treatment and uncolonized plants in the control in 2020 (Mann–Whitney test).

Group	Month	Treatment		U	*p*
		C	Mr		
Microarthropods	June	612 [574; 1339] *	612 [340; 1190]	10.50	0.75
	July	8058 [7089; 10,608]	9180 [5610; 27,030]	7.00	1.00
Collembola	June	0 [0; 128]	0 [0; 0]	9.00	0.53
	July	4284 [3060; 4488]	5610 [1020; 17,340]	6.00	0.77
Oribatida	June	204 [191; 319]	340 [255; 357]	9.50	0.60
	July	204 [102; 306]	510 [0; 3060]	5.00	0.55
Mesostigmata	June	128 [51; 128]	0 [0; 51]	5.00	0.14
	July	829 [612; 867]	1020 [510; 2040]	5.00	0.55
Astigmata	June	127 [127; 306]	170 [102; 170]	11.50	0.92
	July	1683 [1275; 2397]	3060 [510; 3570]	7.00	1.00
Prostigmata	June	51 [51; 127]	0 [0; 51]	7.50	0.35
	July	918 [867; 1084]	1530 [0; 2040]	5.00	0.55

* Data are shown as ME [25; 75], where ME—median; 25 and 75—first and third quartiles, respectively. C—control, Mr—*Metarhizium robertsii*.

**Table 5 insects-13-00807-t005:** Results of comparative analysis of the proportion of individual groups in the community in the plots with different types of seed treatment in 2019 (Mann–Whitney test).

Group	Month	Treatment		U	*p*
		C	Mr		
Collembola	June	7.7 [6.4; 10.9] *	9.8 [9.6; 11.1]	4.0	0.25
	July	0.6 [0.1; 1.5]	0.6 [0.4; 1.2]	8.0	1.00
	August	18.3 [12.2; 29.1]	22.7 [18.9; 25,4]	6.0	0.56
Oribatida	June	55.2 [45.6; 61.8]	49.7 [39.7; 61.2]	8.0	1.00
	July	0.9 [0.6; 1.4]	1.0 [0.6; 1.8]	7.0	0.77
	August	14.3 [7.7; 20.9]	4.7 [2.4; 6.4]	3.0	0.15
Mesostigmata	June	12.3 [7.1; 17.7]	15.0 [9.8; 21.5]	5.0	0.39
	July	0.6 [0.1; 1.9]	0.3 [0.2; 0.4]	8.0	1.00
	August	7.1 [6.5; 8.4]	3.0 [2.2; 4.6]	2.0	0.08
Astigmata	June	22.7 [17.3; 26.2]	15.8 [12.2; 19.8]	5.0	0.39
	July	97.4 [95.4; 98.3]	97.8 [95.8; 98.9]	7.0	0.77
	August	47.5 [39.1; 58.3]	75.8 [69.1; 72.4]	4.0	0.25
Prostigmata	June	5.0 [4.2; 5.7]	3.2 [3.1; 5.7]	7.0	0.77
	July	0.2 [0.1; 0.4]	0.2 [0.0; 0.6]	8.0	1.00
	August	4.0 [2.5; 6.5]	0.8 [0.3; 2.4]	3.0	0.15

* Data are shown as ME [25; 75], where ME—median; 25 and 75—first and third quartiles, respectively. C—control, Mr—*Metarhizium robertsii*.

**Table 6 insects-13-00807-t006:** Results of comparative analysis of the proportion of individual groups in the community in the plots with different types of seed treatment in 2020 (Mann–Whitney test).

Group	Month	Treatment		U	*p*
		C	Mr		
Collembola	June	0.0 [0.0; 6.9] *	0.0 [0.0; 3.5]	12.0	1.00
	July	55.7 [40.3; 61.8]	51.5 [39.6; 63.4]	12.0	1.00
	August	6.1 [5.7; 9.4]	11.0 [5.2; 13.4]	10.0	0.68
Oribatida	June	33.3 [16.7; 34.5]	58.3 [47.1; 60.0]	5.5	0.18
	July	1.9 [1.5; 3.8]	2.3 [1.8; 4.5]	8.5	0.47
	August	0.3 [0.3; 0.6]	0.9 [0.3; 1.1]	6.0	0.21
Mesostigmata	June	20.0 [8.3; 25.0]	6.9 [0.0; 8.3]	3.0	0.06
	July	8.6 [8.2; 9.4]	6.0 [4.9; 6.8]	6.0	0.21
	August	2.4 [2.0; 3.2]	1.9 [1.2; 4.3]	11.0	0.84
Astigmata	June	24.1 [20.0; 50.0]	20.7 [18.2; 40.0]	12.0	1.00
	July	24.3 [15.8; 33.8]	32.8 [13.3; 39.2]	12.0	1.00
	August	87.5 [86.3; 91.3]	80.9 [79.7; 92.6]	10.0	0.68
Prostigmata	June	8.3 [6.9; 25.0]	0.0 [0.0; 8.3]	8.0	0.40
	July	9.9 [8.7; 10.8]	7.3 [4.7; 14.2]	10.0	0.68
	August	2.3 [0.7; 2.9]	1.0 [0.9; 1.9]	9.0	0.53

* Data are shown as ME [25; 75], where ME—median; 25 and 75—first and third quartiles, respectively. C—control, Mr—*Metarhizium robertsii*.

**Table 7 insects-13-00807-t007:** Results of comparative analysis of the proportion of individual groups of soil microarthropods in the communities in the plots taking into account positive plant colonization by the fungus in 2020 (Mann–Whitney test).

Group	Month	Treatment		U	*p*
		C	Mr		
Collembola	June	0.0 [0.0; 7.1] *	0.0 [0.0; 0.0]	10.00	0.68
	July	55.7 [40.3; 61.8]	61.1 [18.2; 64.2]	7.00	1.00
Oribatida	June	33.3 [14.3; 35.7]	47.1 [28.6; 58.3]	8.50	0.46
	July	1.9 [1.5; 3.8]	9.1 [0.0; 11.3]	5.00	0.55
Mesostigmata	June	22.2 [9.5; 25.0]	0.0 [0.0; 8.3]	5.50	0.17
	July	8.2 [7.3; 8.6]	9.1 [7.6; 11.1]	5.00	0.55
Astigmata	June	25.0 [22.2; 50.0]	25.0 [16.7; 50.0]	11.50	0.92
	July	24.3 [15.8; 33.8]	11.3 [5.6; 63.6]	6.00	0.77
Prostigmata	June	8.3 [7.1; 25.0]	0.0 [0.0; 8.3]	8.50	0.46
	July	9.6 [8.7; 10.8]	5.7 [0.0; 22.2]	5.00	0.55

* Data are shown as ME [25; 75], where ME—median; 25 and 75—first and third quartiles, respectively. C—control, Mr—*Metarhizium robertsii*.

## Data Availability

Raw data generated in this study are available upon request.

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
