# Peer review of "The Effect of Bean Seed Treatment with Entomopathogenic Fungus *Metarhizium robertsii* on Soil Microarthropods (Acari, Collembola)"

_insects, 2022, doi:10.3390/insects13090807_

Round 1
Reviewer 1 Report
Novgorodova et al. conducted a comparison between planting using M. robertsii dressed seeds and the normal seeds. The results are interesting that no significant differences were detected on the microarthropod community. And I have several questions:
1. The authors used 5 × 107 conidia/ml conidia to dress the seeds? Is this a standard concentration to do so? As an experiment, why didn’t the authors should try a relative higher concentration in science study to see the potential influence even under an un-predicted high concentration?
2. The methods of the authors to collect the microarthropods seemed to be “standard”, however, I am worried about the situation of missing some individuals or leave them die in the dry soil. So can the author explain why this method is reasonable?
3. What’s the detailed changes inner each clade, for example, insects. Because there might be an increase for order A but decrease in order B, finalizing in a balanced number. So can the authors provide more clues?
4. The authors should supplement more description on how the dressed seeds protected the plant from pests.
Author Response
Dear Reviewer,
We apologize for the inconvenience. Apparently, during the submitting process, the program at some point partially translated the page.
Thanks a lot for your helpful discussion of the results and valuable comments on the manuscript. The changes within the m/s are highlighted. The answers to your comments are below.
Best regards,
Natalia Vladimirova et al.
Response to Reviewer 1 Comments
Extensive editing of English language and style required.
Response: English language of the manuscript was improved by the native speaker.
Point 1: The authors used 5 × 107 conidia/ml conidia to dress the seeds? Is this a standard concentration to do so? As an experiment, why didn’t the authors should try a relative higher concentration in science study to see the potential influence even under an un-predicted high concentration?
Response 1: The following phrase was added: ‘Using this concentration of fungal conidia suspension is optimal, since it makes it possible to obtain a positive effect on plant growth, reduces the infection level of the seed material with phytopathogens, and significantly decreases the development and prevalence of root rot disease [4, 22, 24, 25, 26].’
The references are: Sasan, Bidochka, 2012; Ahmad et al., 2020; Quesada-Moraga, 2020; Tomilova et al., 2020; Ashmarina et al., 2021.
Point 2: The methods of the authors to collect the microarthropods seemed to be “standard”, however, I am worried about the situation of missing some individuals or leave them die in the dry soil. So can the author explain why this method is reasonable?
Response 2: In order to prevent expedited soil drying and missing some individuals we used modified Berlese funnels without additional sources of light and heat. ‘This way of microarthropod extraction provides natural conditions for soil drying and avoids or at least minimizes the possibility of missing some individuals due to the expedited drying process.’ To emphasize this, the last phrase was added to the text.
Point 3: What’s the detailed changes inner each clade, for example, insects. Because there might be an increase for order A but decrease in order B, finalizing in a balanced number. So can the authors provide more clues?
Response 3: Unfortunately, the question is not quite clear, since insects are not mentioned in the manuscript at all. Our study was focused on effect of endophytic fungus M. robertsii on soil microarthropods as one of the key ‘soil-makers’
As for a lack of information about community structure of Collembola and the studied groups of Acari in the manuscript, we did not plan to make a detailed analysis of these groups in this study. The role of microarthropods (in total) in soil formation, accelerating the decomposition and humification of organic residues is known to be high (Wagg et al., 2014; Coleman et al., 2017). The decrease of microarthropod abundance is supposed to have a negative effect on soil fertility and ultimately on yields. However, the role of separate species in soil formation is still unclear (Behan-Pelletier, 2003; Kampichler, 2009). That is why, at the first stage we focused on the abundance and community structure (using only key groups) of soil microarthropods to highlight the problem in a whole.
Anyway, it is a very interesting, though not so simple, question, and we plan to address it in our future research.
Point 4: The authors should supplement more description on how the dressed seeds protected the plant from pests.
Response 4: This study was not directly focused on the plant protection from pests. However, in the previous paper presenting data on the plant colonisation with fungus it was shown that treatment of the bean seeds with M. robertsii significantly reduced the development and prevalence of root rot disease (Ashmarina et al., 2022). This information was added to the first paragraphs of Results and Discussion.
----------------------------------------------------------------------------------------
Dear reviewer,
Thanks a lot for your helpful discussion of the results and valuable comments on our manuscript. The changes within the ms are highlighted. The answers to your comments are below.
Best regards, Natalia Vladimirova et al.
Response to Reviewer 1 Comments
Extensive editing of English language and style required.
Response: English language of the manuscript was improved by the native speaker.
Пункт 1: Авторы использовали 5 × 107 конидий / мл конидий для одевания семян? Является ли это стандартной концентрацией для этого? В качестве эксперимента, почему авторы не должны попробовать относительно более высокую концентрацию в научном исследовании, чтобы увидеть потенциальное влияние даже при непрогнозированной высокой концентрации?
Ответ 1: Добавлена следующая фраза: «Использование такой концентрации суспензии грибковых конидий оптимально, так как позволяет получить положительный эффект на рост растений, снижает уровень инфицирования семенного материала фитопатогенами, значительно снижает развитие и распространенность болезни корневой гнили [4, 22, 24, 25, 26]».
Ссылки: Сасан, Бидочка, 2012; Ахмад и др., 2020; Кесада-Морага, 2020; Томилова и др., 2020; Ашмарина и др., 2021.
Пункт 2: Методы авторов по сбору микробронходов казались «стандартными», однако, меня беспокоит ситуация исчезновения некоторых особей или оставления их умирать в сухой почве. Так может ли автор объяснить, почему этот метод разумен?
Ответ 2: Чтобы предотвратить ускоренное высыхание почвы и отсутствие некоторых особей, мы использовали модифицированные воронки Берлеза без дополнительных источников света и тепла. «Этот способ экстракции микропартропода обеспечивает естественные условия для сушки почвы и позволяет избежать или, по крайней мере, свести к минимуму возможность пропуска некоторых особей из-за ускоренного процесса сушки». Чтобы подчеркнуть это, в текст была добавлена последняя фраза.
Пункт 3: Какие детальные изменения внутри каждой клады, например, насекомые. Потому что может быть увеличение для порядка А, но уменьшение в порядке В, завершение в сбалансированном числе. Так могут ли авторы дать больше подсказок?
Ответ 3: К сожалению, вопрос не совсем ясен, так как насекомые в рукописи вообще не упоминаются. Наше исследование было сосредоточено на влиянии эндофитного гриба M. robertsii на почвенные микропартроподы как одного из ключевых «почвообразователей»
Что касается отсутствия в рукописи информации о структуре сообщества Коллембола и исследуемых группах Акари, то мы не планировали делать детальный анализ этих групп в данном исследовании. Известно, что роль микроартроподов (в общей сложности) в почвообразовании, ускоряющем разложение и гумификацию органических остатков ( Wagg et al., 2014; Coleman et al., 2017). Предполагается, что уменьшение численности микроэлементов отрицательно сказывается на плодородии почв и, в конечном счете, на урожайности. Однако роль отдельных видов в почвообразовании до сих пор неясна (Behan-Pelletier, 2003; Кампичлер, 2009). Именно поэтому на первом этапе мы сосредоточились на численности и структуре сообществ (используя только ключевые группы) почвенных микробронходов, чтобы выделить проблему в целом.
Во всяком случае, это очень интересный, хотя и не такой простой, вопрос, и мы планируем рассмотреть его в наших будущих исследованиях.
Пункт 4: Авторы должны дополнить более подробное описание того, как заправленные семена защищали растение от вредителей.
Ответ 4: Это исследование не было непосредственно сосредоточено на защите растений от вредителей. Однако в предыдущей работе, представляющей данные о колонизации растений грибком, было показано, что обработка семян фасоли M. robertsii значительно снижала развитие и распространенность болезни корневой гнили (Ashmarina et al., 2022). Эта информация была добавлена в первые пункты раздела «Результаты и обсуждение».
Reviewer 2 Report
The article presented for review presents the results of experiments on the influence of entomopathogenic fungi (Metarhizium robertsii) on some soil arthropods (Acari, Collembola). Overall, the article was well written. However, a cursory design of the experiments made it impossible to obtain more scientifically valuable results. The only conclusion that can be drawn from the conducted experiments it the lower abundance of microatrhropods. However, even this conclusion should be treated with great caution. In my opinion, the data should not have been combined before analysis.
More important comments:
L25: entomopathogenic fungi rather do not stimulate plant growth.
L39: method of what? Plant protection?
L47-48: check grammar.
L96: why twice?
L109: explain the method of root washing; why 1 min at 180 rpm?
L119: remove „quite”.
L128 (relative humidity), data presented in the table 1: without knowledge about the composition of the soil, its osmotic potential, the presented data on moisture content is useless.
L175-185: The paragraph basically applies in its entirety to unpublished data presented elsewhere. This is unacceptable.
L188-198: the Shapiro-Wilk test is only a preliminary test that precedes further statistical analysis. The results of this test are not presented.
L206: if the increase was not statistically significant, it was not an increase.
Table 2 and next: use p>0.05. Do not give exact p values. They are not statistically significant anyway. Why did the authors use ME? Why not standard deviation or standard error?
L242: The size of a group does not necessarily determine its importance in the community.
L258-263: The paragraph basically applies in its entirety to unpublished data presented elsewhere. This is unacceptable.
L290-306: there is no citations.
L320-322: were the data statistically compared? If not, the differences are not significant.
Author Response
Dear reviewer,
Thanks a lot for your helpful discussion of the results and valuable comments on our manuscript. The changes within the ms are highlighted. The answers to your comments are below.
Best regards, Natalia Vladimirova et al.
Response to Reviewer 2 Comments
The article presented for review presents the results of experiments on the influence of entomopathogenic fungi (Metarhizium robertsii) on some soil arthropods (Acari, Collembola). Overall, the article was well written. However, a cursory design of the experiments made it impossible to obtain more scientifically valuable results. The only conclusion that can be drawn from the conducted experiments it the lower abundance of microatrhropods. However, even this conclusion should be treated with great caution. In my opinion, the data should not have been combined before analysis.
Response: We respect your opinion, but we cannot agree that the design of the experiments was cursory. The experiments were carried out in accordance with the field research standards and the goals of the study.
No adverse effects of the bean seed treatment with the entomopathogenic fungus M. robertsii on the abundance and community structure of soil microarthropods were revealed in two-year field experiment.
No data were combined before analysis. Our study was focused on effect of endophytic fungus M. robertsii on soil microarthropods as one of the key ‘soil-makers’. The role of microarthropods (in total) in soil formation, accelerating the decomposition and humification of organic residues is known to be high (Wagg et al., 2014; Coleman et al., 2017). The decrease of microarthropod abundance is supposed to have a negative effect on soil fertility and ultimately on yields. However, the role of separate species in soil formation is still unclear (Behan-Pelletier, 2003; Kampichler, 2009). That is why, at the first stage we focused on the abundance and community structure of soil microarthropods to highlight the problem in a whole. So, detailed analysis of community structure of Collembola and the studied groups of Acari was not planned in the study.
Point 1: L25: entomopathogenic fungi rather do not stimulate plant growth.
Response 1: In fact, they do stimulate plant growth, that was shown experimentally (Sasan, Bidochka, 2012; Ashmarina et al., 2022). However, since until recently this was not the main goal when dressing seeds with pathogenic fungi the phrase was changed as follows: ‘Therefore, dressing seeds with a conidial suspension for plant inoculation, with entomopathogenic fungi (at least M. robertsii), can be assumed as a potentially safe method of plant protection for soil microarthropods.’
Point 2: L39: method of what? Plant protection?
Response 2: The phrase was changed as follows: ‘The results indicate the possibility of using dressing seeds with conidial suspension for plant inoculation with entomopathogenic fungus (at least M. robertsii) as a potentially safe plant protection method for non-target soil microarthropods.’
Point 3: L47-48: check grammar.
Response 3: Grammar of the ms was checked by a native English-speaking colleague.
Point 4: L96: why twice?
Response 4: The following phrase was added: ‘Double autoclaving ensured the sterility of the substrate’.
Point 5: L109: explain the method of root washing; why 1 min at 180 rpm?
Response 5: This part was changed as follows: ‘To analyse rhizosphere colonization, the pre-washed roots of the plants were plated without surface sterilization. After preliminary washing of a dense layer of basal soil on bean plants (at least twice), the roots were washed three times for 1 minute using Vortex-Mixer PV-1 (Grant-bio) and plated on the medium in Petri dishes.’
Point 6: L119: remove „quite”.
Response 6: It was done. ‘Treatment of planting material by dressing seeds with conidial suspension is a typical method used for plant inoculation [4, 22, 24, 25].’
Point 7: L128 (relative humidity), data presented in the table 1: without knowledge about the composition of the soil, its osmotic potential, the presented data on moisture content is useless.
Response 7: We can’t agree. Temperature and relative humidity of soil are quite important factors that may have strong effect on the microarthropod abundance. Estimating of these parameters is obligatory part of any experiments that concern microarthropod communities, especially field experiments (e.g., Andrewsa, Ruess, 2020). These data are required to adequate analysis of any results obtained in the field.
Andrewsa, R.N.; Ruess, R.W. Microarthropod abundance and community structure along a chronosequence within the Tanana River floodplain, Alaska. ÉCOSCIENCE 2020, 27, 235-253. https://doi.org/10.1080/11956860.2020.1772613.
Point 8: L175-185: The paragraph basically applies in its entirety to unpublished data presented elsewhere. This is unacceptable.
Response 8: The data are available in preprint (Ashmarina et al., 2022). The reference was changed in the list.
Point 9: L188-198: the Shapiro-Wilk test is only a preliminary test that precedes further statistical analysis. The results of this test are not presented.
Response 9: In this case, Shapiro–Wilk test was used to estimate the distribution of microarthropods (both in general and their individual groups) in the experimental field before the start of the experiments, so no additional analysis using other tests is required.
The following phrase was added to the section 2.4. Data analysis: ‘To get preliminary information on the distribution of microarthropods (both in general and their individual groups) in the field before the start of the experiments, the Shapiro–Wilk test was used. P values less than 0.05 meant uneven distribution.’
Point 10: L206: if the increase was not statistically significant, it was not an increase.
Response 10: We can’t agree with this statement: the word increase does not mean statistical significance by itself and require statistical analysis that was done in the study. In this case, we have to comment the visible in the Figure increase of microarthropod abundance. The results presented in the manuscript clearly show that visible differences are non-significant.
Point 11: Table 2 and next: use p>0.05. Do not give exact p values. They are not statistically significant anyway.
Why did the authors use ME? Why not standard deviation or standard error?
Response 11: We would prefer to give the exact p values, as they reflect the real situation, e.g., p = 0.06 is quite far from p = 0.9 and is worth mentioning. In addition, it may be useful in planning future experimental studies. However, if you insist, of course we can use p>0.05 instead of the exact values.
Because of the non-normal distribution of microarthropods in most cases (Shapiro–Wilk test, p < 0.05), the influence of plant seed treatment on both the abundance of microarthropods and the structure of their communities (the proportion of individual groups) was estimated using the non-parametric test (Mann-Whitney test). In this case, usage of mean values and standard deviation or standard error makes no sense.
Point 12: L242: The size of a group does not necessarily determine its importance in the community.
Response 12: Thanks a lot for your question. The phrase was changed as follows: ‘The proportion of Prostigmata in the community throughout the season did not exceed 5%.’
Point 13: L258-263: The paragraph basically applies in its entirety to unpublished data presented elsewhere. This is unacceptable.
Response 13: The data are available in preprint (Ashmarina et al., 2022). The reference was changed in the list.
Point 14: L290-306: there is no citations.
Response 14: This concerns only data obtained in this study, so no citations could be added.
Point 15: L320-322: were the data statistically compared? If not, the differences are not significant.
Response 15: Thanks a lot for your question. The phrase was changed as follows: ‘The abundance of both Mesostigmata and Prostigmata also increased in the middle of the season; however, the increase was not pronounced, and their proportion in the community was 2-6 times lower than the proportion of the first two groups in the middle of the season.
Round 2
Reviewer 1 Report
I cannot understand the language.
Author Response
Dear Reviewer,
We apologize for the inconvenience. Apparently, during the submitting process, the program at some point partially translated the page.
Thanks a lot for your helpful discussion of the results and valuable comments on the manuscript. The changes within the m/s are highlighted. The answers to your comments are below.
Best regards,
Natalia Vladimirova et al.
Reviewer 2 Report
Accept in present form.
Author Response
Dear Reviewer,
Thanks a lot for your helpful discussion of the results and valuable comments on the ms.
Best regards,
Natalia Vladimirova et al.
Round 3
Reviewer 1 Report
Agree to publish.
Author Response

(The authors gave the same response as above.)
